# REDUCING THE TEACHER-STUDENT GAP VIA ADAPTIVE TEMPERATURES

## ABSTRACT

Knowledge distillation aims to obtain a small and effective deep model (student) by learning the output from a larger model (teacher). Previous studies found a severe degradation problem, that student performance would degrade unexpectedly when distilled from oversized teachers. It is well known that larger models tend to have sharper outputs (Guo et al., 2017). Based on this observation, we found that the sharpness gap between the teacher and student output may cause this degradation problem. To solve this problem, we first propose a metric to quantify the sharpness of the model output. Based on the second-order Taylor expansion of this metric, we propose Adaptive Temperature Knowledge Distillation (ATKD), which automatically changes the temperature of the teacher and the student, to reduce the sharpness gap $G_{s\_gap}$. We conducted extensive experiments on CIFAR100 and ImageNet and achieved significant improvements. Specifically, ATKD trained the best ResNet18 model on ImageNet as we knew (73.0% accuracy).

## 1 INTRODUCTION

Deep neural networks have achieved remarkable success in most of computer vision tasks (He et al., 2016; Deng et al., 2009), while the increasing network capacity in the current state-of-the-art models also results in severe computational burdens and high inference time (Tian et al., 2020; Yim et al., 2017). One direction to reduce the model size is knowledge distillation (KD), which trains small efficient models (student) by learning from the output of large models (teacher) (Tang et al., 2020). In 2015, Hinton proposed to soften the model output by raising the temperature of the last softmax layer(Hinton et al., 2015). Since then, this temperature-based kowledge distillation has drawn the main stream attentions and achieved many successes (Zagoruyko & Komodakis, 2017; Kim et al., 2020; Furlanello et al., 2018; Fu et al., 2020).

However, recent researches found that knowledge distillation suffers from a mysterious performance degradation problem(Cho & Hariharan, 2019; Mirzadeh et al., 2019). Specifically, since the idea of KD is transferring the knowledge of teacher into student, a nature assumption is larger teachers would train better students. But recent researches refuted this assumption and found that the student often performs worse with oversized teachers(Cho & Hariharan, 2019).

This paper studies this degradation problem by considering the sharpness (or softness) of the model output. It is well known that networks with more parameters tend to produce sharper output (Guo et al., 2017; Lee et al., 2018), which would create the gap of sharpness between teacher and student, i.e. the teacher is usually much sharper than the student. It should be noted that temperature technology in vanilla knowledge distillation does not reduce this gap. This is because in vanilla knowledge distillation, the teacher and student are softened with the same temperature, which maintains the large gap of sharpness. When the teacher grows larger, this gap will further increases, making it more difficult for the student to learn from the teacher.

To solve this problem, let us consider a modified softer teacher model with the same accuracy. In this way, the sharpness gap between the student and the teacher models would be narrowed, making learning more accessible. One way to build this softer teacher is to use higher temperatures for the teacher model, but it would require labor-intensive searches on the validation data set.

With this insight, we propose to adaptively change the temperatures of the teacher and the student by their sharpness. Our contribution is three-fold:

1. We propose a metric to quantify the sharpness of models, allowing us to precisely control the sharpness of model output. Formally, we use the realsoftmax function (Nielsen & Sun, 2016) on the logits vector as the sharpness metric, which approximates a smoother version of the max function. If we denote student logits as $z$, the teacher logits as $v$, teacher temperature and student temperature as $\tau^T$ and $\tau^S$ respectively ($\tau^T$ is equal to $\tau^S$ in the vanilla knowledge distillation), the sharpness metric and the gap of sharpness are defined as follows:

$$
\begin{aligned}
S_{sharpness} &= \log \sum_j e^{z_j/\tau} \\
G_{s\_gap} &= \log \sum_j e^{v_j/\tau^T} - \log \sum_j e^{z_j/\tau^S}
\end{aligned}
\tag{1}
$$

2. We propose Adaptive Temperature Knowledge Distillation (ATKD), which automatically changes the temperatures of the teacher and the student, to reduce the sharpness gap $G_{s\_gap}$. ATKD relieves the burden of searching for the appropriate temperature on the validation set. Specifically, by Taylor second-order expansion, our method can be implemented easily by normalizing the logits with the standard deviation.

3. Our $S_{sharpness}$ metric can be used to explain why two existing methods work. Specifically, the Early Stopped method and the Teacher Assistant method both reduce the sharpness gap between the teacher and the student.

We present comprehensive experiments on CIFAR-100 Krizhevsky et al. (2009) and ImageNet datasets (Deng et al., 2009) to evaluate our method. The experiment results show that: 1) The proposed ATKD method tremendously mitigates the performance degradation problem and trains the best ResNet18 model on ImageNet as we know (73.01% accuracy). 2) The proposed ATKD method is easy to optimize, exhibits much lower training/test loss with oversized teachers than KD.

## 2 METHODOLOGY

### 2.1 BACKGROUND

**Vanillay Knowledge Distillation** During training, we minimize the negative log likelihood of the ground truth class to update model parameters. After the model is properly trained, the probability of the ground truth would be close to 1, while the rest wrong predictions are near zeros.

In 2015, Hinton noticed that these small wrong probabilities are useful to unveil "dark knowledge" (Hinton et al., 2015). Take a picture of "cat" for example, the model are more likely to output higher probability for class "dog" than class "airplane". These wrong probabilities imply the relationship between the two classes and unveil how a model tends to generalize. This observation inspired to use the output of large models as soft targets to train efficient small models.

However, modern deep networks tend to produce peaky probabilities (Guo et al., 2017; Lee et al., 2018), that the numbers of those wrong classes (near zero values) would be negligible compared to the ground truth (near one). Thus Hinton proposed to raise the temperature of the last softmax to soften the output probabilities, which can be used as soft targets to train small networks.

We denote logits of the teacher as $v$, and logits of the student as $z$, temperature as $\tau$, i and j denote the ith and jth value of logits (i.e. the ith and jth category of K classes). the loss of knowledge distillation is as follows:

$$
\begin{aligned}
\mathcal{L}_{KD} &= -\sum_i p_i^T \log p_i^S \\
p_i^S &= \frac{e^{z_i/\tau}}{\sum_j e^{z_j/\tau}}, p_i^T = \frac{e^{v_i/\tau}}{\sum_j e^{v_j/\tau}}
\end{aligned}
\tag{2}
$$

The final loss for the student is then the weighted sum of the cross entropy loss $\mathcal{L}_{cls}$ and the knowledge distillation loss $\mathcal{L}_{KD}$:

$$
\mathcal{L} = \lambda \mathcal{L}_{KD} + (1 - \lambda)\mathcal{L}_{cls}
\tag{3}
$$

Table 1: The Performance Degradation Problem.

| Teacher | ResNet20 | ResNet32 | ResNet44 | ResNet56 | ResNet110 |
|---|---|---|---|---|---|
| Teacher Acc | 69.57 | 70.9 | 71.9 | 72.8 | **73.8** |
| Student Acc | 67.4 | **68.2** | 68 | 67.5 | 67.1 |
| KD loss | **1.1** | 1.7 | 2.1 | 2.5 | 3.3 |

The popular choice of the temperature $\tau$ is in $\{3, 4, 5\}$ and the weight $\lambda = 0.9$ (Hinton et al., 2015; Cho & Hariharan, 2019; Tian et al., 2020).

**Performance Degradation Problem** While knowledge distillation achieved success in many fields, a mysterious performance degradation problem was spotted in 2019. Since the idea of knowledge distillation is transferring teacher knowledge into students, one natural hypothesis is that a larger and more accurate teacher would capture more knowledge and thus train better students. Unfortunately, previous studies invalidate this hypothesis by showing that the student performance would degenerate unexpectedly with larger teachers.

Table 1 shows our experiment result. The student is ResNet14. With larger teachers, the student accuracy degrades, and the KD loss increase. Cho & Hariharan (2019) hypothesizes that the mismatch of capacity causes this problem. We will discuss this problem in the rest of the section.

## 2.2 Adaptive Temperature Knowledge Distillation

We investigated the degradation problem and proposed to use adaptive temperatures during training to mitigate this problem:

1. We found that the mismatch sharpness between the teacher and the student may cause the degradation problem.

2. We propose a metric to quantify the sharpness of a model.

3. We propose the adaptive temperature knowledge distillation (ATKD) methods.

**The sharpness of neural networks** Two main theories explain why knowledge distillation is effective. The first theory originated from Hinton's first paper, which argued that teachers are more accurate in capturing category similarities. This information of category similarity helps students to generalize better on unseen data. On the other hand, some contend that the soft output of the teacher prevents the student network from overconfidence (ie. label smoothing technique).

However, when the teacher gets larger, these two theories will contradict each other. This is because that larger models are usually more accurate and peaky at the same time. On the one hand, a more accurate teacher is beneficial to the student model. On the other hand, a sharper teacher tends to make the student overconfident. Therefore, the larger teacher capacity is a double-edged sword for distillation. This analysis shows that the sharpness of teachers should be investigated carefully, while previous studies have overlooked this critical attribute of teachers.

To this end, we propose to use realsoftmax (Nielsen & Sun, 2016) to quantify the sharpness of the model output, which is defined as the logarithm of the sum of the exponentials of the logits:

$$S_{sharpness} = \log \sum_j e^{z_j} \tag{4}$$

Such metric has at least two advantages, therefore reasonably measure the sharpness of the model output: 1) This metric is a smooth approximation to the maximum function $max_j(z)$ 2) It is also differentiable.

Meanwhile, we quantify the sharpness gap between the teacher and the student as the difference of the sharpness metric:

$$G_{s\_gap} = \log \sum_j e^{v_j} - \log \sum_j e^{z_j} \tag{5}$$

Table 2: The $S_{sharpness}$ of different models with temperature set to one.

| Network | ResNet14 | ResNet20 | ResNet32 | ResNet44 | ResNet56 | ResNet110 |
|---|---|---|---|---|---|---|
| $S_{sharpness}$ | 12.20 | 13.11 | 13.84 | 14.45 | 15.37 | 16.13 |

If we consider the temperatures, the sharpness gap is:

$$G_{s\_gap} = \log \sum_j e^{v_j/\tau} - \log \sum_j e^{z_j/\tau} \tag{6}$$

Table 2 shows sharpness metric value for models of different capacity. It shows that $S_{sharpness}$ increases with network capacity, and there is a sharpness gap between networks of different sizes.

**Adaptive Temperature** One idea is using higher temperatures for the teacher to reduce the sharpness gap between teachers and students. However, searching on the validation data set requires a lot of computational costs. Instead, we propose adaptive temperature knowledge distillation, which automatically tunes the temperature according to the $S_{sharpness}$.

By Taylor second expansion:

$$\begin{aligned}
G_{s\_gap} &= \log \sum_j e^{v_j/\tau} - \log \sum_j e^{z_j/\tau} \\
&\approx \log(K + \sum_j v_j/\tau + \frac{1}{2} \sum_j v_j^2/\tau^2) \\
&\quad - \log(K + \sum_j z_j/\tau + \frac{1}{2} \sum_j z_j^2/\tau^2)
\end{aligned} \tag{7}$$

We follow the assumption from Hinton (Hinton et al., 2015), that the logits have been zero-meaned separately for each training example so that $\sum_j z_j = \sum_j z_j = 0$. We found from experiments that the sum of logits is indeed very small numbers close to zero. The experiment results and further theoretical analysis are provided in the appendix.

Given the above assumption $\sum_j z_j = \sum_j z_j = 0$, We can get:

$$\begin{aligned}
G_{s\_gap} &= \log(K + \frac{1}{2} \sum_j (v_j/\tau)^2) - \log(K + \frac{1}{2} \sum_j (z_j/\tau)^2) \\
&= \log(1 + \frac{1}{2\tau^2 K} \sum_j v_j^2 - \log(1 + \frac{1}{2\tau^2 K} \sum_j z_j^2))
\end{aligned} \tag{8}$$

The $\frac{1}{K} \sum_j v_j^2$ in the equation is the variance of logits, which can be represented by the square of the standard deviation $std^2$:

$$G_{s\_gap} = \log(1 + \frac{1}{2} * (std^T/\tau)^2) - \log(1 + \frac{1}{2} * (std^S/\tau)^2) \tag{9}$$

This result shows the relationship between the temperature and sharpness gap is pretty straightforward, that the standard deviation controls the sharpness gap between the teacher and the student. Therefore, we set the temperature of the teacher and the student as the standard deviation of the logits vector respectively. These adaptive temperatures would automatically change during training, reducing the sharpness gap between the teacher and the student. It is worth noting that there is no need to search for the proper temperature on the validation set like the vanilla KD.

Formally, if we denote standard deviation function as $Std$, we propose Adaptive Temperature Knowledge Distillation as follows:

$$\tau_i^T = Std(v_i), \tau_i^S = Std(z_i)$$

$$\mathcal{L}_{ATKD} = -\sum_i p_i^T \log p_i^S$$

$$p_i^S = \frac{e^{z_i/\tau_i^S}}{\sum_j e^{z_j/\tau_i^S}}, p_i^T = \frac{e^{v_i/\tau_i^T}}{\sum_j e^{v_j/\tau_i^T}}$$

$$\mathcal{L} = \lambda\mathcal{L}_{ATKD} + (1-\lambda)\mathcal{L}_{cls}$$

(10)

### 2.3 Sharpness Gap Analysis

We investigated the relationship between the sharpness gap and the temperature, and found that the $G_{s\_gap}$ of ATKD decrease with $\frac{1}{\tau^3}$, while the vanilla KD decrease with $\frac{1}{\tau^2}$. Therefore, If the temperature of KD and adaptive temperatures of ATKD are in the same temperature range, the sharpness gap of ATKD $G_{s\_gap}$ would be smaller than KD in a considerable margin.

If we use $K$ to denote $K$ classes, the sharpness gap would decrease with $\frac{1}{\tau^2}$ under vanilla KD:

$$G_{s\_gap} = \log \sum_j e^{v_j/\tau} - \log \sum_j e^{z_j/\tau}$$

$$= \log(1 + O(\frac{1}{\tau^2})) - \log(1 + O(\frac{1}{\tau^2})) + logK - logK$$

$$\approx O(\frac{1}{\tau^2})$$

(11)

This shows that when the temperature is very high and the linear function $(1+x)$ can approximate $e^x$, $G_{s\_gap}$ would be reduced. However, with this linear approximation, the KD loss would degenerate into MSE loss, as Hinton points out (Hinton et al., 2015):

$$\frac{\partial L}{\partial z_i} = \frac{1}{T}\left(\frac{e^{z_i/\tau}}{\sum_j e^{z_j/\tau}} - \frac{e^{v_i/\tau}}{\sum_j e^{v_j/\tau}}\right)$$

(12)

$$\approx \frac{1}{T}\left(\frac{z_i/\tau + 1}{K} - \frac{v_i/\tau + 1}{K}\right)$$

(13)

$$\approx \frac{1}{K\tau^2}(z_i - v_i)$$

(14)

However, high temperatures could be harmful to student, because it would encourage the student to learn more from those negative logits of the teacher (Hinton et al., 2015), and as Hinton pointed out, these negative values of logits could be very noisy.

On the other hand, the sharpness gap of ATKD would decrease faster with $\frac{1}{\tau^3}$, and enable the ATKD to achieve much lower sharpness gap than KD. Because we can use second-order Taylor approximation $(1 + x + \frac{1}{2}x^2)$, which is a better approximation than linear function $(1 + x)$ in a wide range.

$$G_{s\_gap} = \log \sum_j e^{v_j/\tau^T} - \log \sum_j e^{z_j/\tau^S}$$

$$= \log(1 + \frac{1}{2}*(std^S/\tau^S)^2 + O((\frac{1}{\tau^S})^3))$$

$$- \log(1 + \frac{1}{2}*(std^T/\tau^T)^2 + O((\frac{1}{\tau^T})^3))$$

$$\approx O((\frac{1}{\tau^T})^3) - O((\frac{1}{\tau^S})^3)$$

(15)

## 3 EXPERIMENTS

In this section, we will show comprehensive experiment results to validate the effectiveness of ATKD from several perspectives. Specifically, we first conducts experiments on two popular CV datasets

Table 3: CIFAR-100 experiments.

| Teacher
Student | WRN-40-2
WRN-16-2 | WRN-40-2
WRN-40-1 | ResNet56
ResNet20 | ResNet110
ResNet20 | ResNet110
ResNet32 | ResNet32*4
ResNet8*4 | VGG13
VGG8 |
|---|---|---|---|---|---|---|---|
| Teacher | 75.61 | 75.61 | 72.34 | 74.31 | 74.31 | 79.42 | 74.64 |
| Student | 73.26 | 71.98 | 69.06 | 69.06 | 71.14 | 72.50 | 70.36 |
| KD | 74.92 | 73.54 | 70.66 | 70.67 | 73.08 | 73.33 | 72.98 |
| FitNet | 73.58 | 72.24 | 69.21 | 68.99 | 71.06 | 73.50 | 71.02 |
| AT | 74.08 | 72.77 | 70.55 | 70.22 | 72.31 | 73.44 | 71.43 |
| SP | 73.83 | 72.43 | 69.67 | 70.04 | 72.69 | 72.94 | 72.68 |
| CC | 73.56 | 72.21 | 69.63 | 69.48 | 71.48 | 72.97 | 70.71 |
| VID | 74.11 | 73.30 | 70.38 | 70.16 | 72.61 | 73.09 | 71.23 |
| RKD | 73.35 | 72.22 | 69.61 | 69.25 | 71.82 | 71.90 | 71.48 |
| PKT | 74.54 | 73.45 | 70.34 | 70.25 | 72.61 | 73.64 | 72.88 |
| AB | 72.50 | 72.38 | 69.47 | 69.53 | 70.98 | 73.17 | 70.94 |
| FT | 73.25 | 71.59 | 69.84 | 70.22 | 72.37 | 72.86 | 70.58 |
| FSP | 72.91 | - | 69.95 | 70.11 | 71.89 | 72.62 | 70.23 |
| NST | 73.68 | 72.24 | 69.60 | 69.53 | 71.96 | 73.30 | 71.53 |
| CRD | 75.48 | 74.14 | 71.16 | 71.46 | 73.48 | 75.51 | 73.94 |
| **ATKD** | **75.75** | **75.06** | **72.08** | **72.12** | **74.09** | **76.40** | **74.17** |

Table 4: ImageNet experiments with Top1 accuracy.

| CE | KD | ES | SP | CC | CRD | AT | ATKD |
|---|---|---|---|---|---|---|---|
| 69.8 | 69.20 | 71.40 | 70.62 | 69.96 | 71.38 | 70.70 | **72.80** |

to demonstrate the performance of ATKD. Then we focused on evaluating whether it could alleviate the performance degradation problem.

**Dataset** 1) *CIFAR-100* (Krizhevsky et al., 2009) is a relatively small data set and is widely used for testing various of deep learning methods. CIFAR-100 contains 50,000 images in the training set and 10,000 images in the dev set, divided into 100 fine-grained categories. 2) *ImageNet* (Deng et al., 2009) is a much larger one than CIFAR-100. ImageNet contains 1.2M images for training and 50K for validation, that distributes in 1000 classes.

**CIFAR Experimental settings** We run a total of 240 epochs for all methods. The learning rate is initialized as 0.05, then decay by 0.1 every 30 epochs after 150 epochs. Temperature is 4 for vanilla KD, and the weight of ATKD or KD and cross-entropy is 0.9 and 0.1 for all the settings.

**ImageNet Experimental settings** Here we use ResNet18 as student for all of methods. Training settings like learning rate or training epochs are the same with Heo et al. (2019) for ImageNet. The teacher network is well-trained previously and fixed during training.

## 3.1 CIFAR-100 AND IMAGENET

**Baselines.** We selected many SOTA KD methods to evaluate the performances of ATKD. Knowledge defined from intermediate layers: FitNet (Romero et al., 2015), AT (Zagoruyko & Komodakis, 2017), SP (Tung & Mori, 2019), PKT (Passalis & Tefas, 2018), FT (Kim et al., 2020), FSP (Yim et al., 2017) . 1) Knowledge defined via mutual information: CC (Peng et al., 2019), VID (Ahn et al., 2019), CRD (Tian et al., 2020). 2) Structured Knowledge: RKD (Park et al., 2019). 3) Knowledge from logits: KD (Hinton et al., 2015), NST (Huang & Wang, 2017), ES (Cho & Hariharan, 2019), TA (Mirzadeh et al., 2019)

**Results in CIFAR-100.** Table 3 shows that ATKD always has an outstanding improvement compared with all other methods. In some situations (e.g. those where teacher/student is WRN-40-2/WRN-40-1 or ResNet110/ResNet32), the performances of ATKD are even very close to those of teacher.

**Results in ImageNet.** All experiments used ResNet34 as the teacher and ResNet18 as the student. Table 4 (the Top1 accuracy) shows that ATKD exceeds all of the previous SOTA by a large margin. Before ATKD, the improvement of this task is limited. Fig. 1 shows the training process of

Table 5: Performance Degradation Problem on CIFAR-100. Student is ResNet14. ATKD archives lower training loss and higher accuracy. The $G_{s\_gap}$ between the distilled student and teacher is also reduced significantly. Temperature is set to 4 in vanilla KD.

|  |  | ResNet20 | ResNet32 | ResNet44 | ResNet56 | ResNet110 |
|---|---|---|---|---|---|---|
| Training loss | Vanilla KD | 1.1 | 1.7 | 2.1 | 2.5 | 3.3 |
|  | ATKD | **0.9** | **1.2** | **1.3** | **1.4** | **1.6** |
| Test acc | Vanilla KD | 67.4 | 68.2 | 68 | 67.5 | 67.1 |
|  | ATKD | **68.2** | **68.7** | **68.9** | **68.8** | **69.2** |
| $G_{s\_gap}$ | Vanilla KD | 0.09 | 0.16 | 0.21 | 0.30 | 0.38 |
|  | ATKD | **0.05** | **0.11** | **0.14** | **0.20** | **0.25** |

Table 6: Performance Degradation Problem on ImageNet.

| Teacher | Method | Accuracy | Teacher | Method | Accuracy |
|---|---|---|---|---|---|
| ResNet34 | KD | 69.43 | ResNet101 | - | - |
|  | ES | 70.98 |  | - | - |
|  | **ATKD** | **72.80** |  | **ATKD** | **72.85** |
| ResNet50 | KD | 69.05 | ResNet152 | - | - |
|  | TA | 70.65 |  | TA | 70.59 |
|  | ES | 70.95 |  | ES | 70.74 |
|  | **ATKD** | **73.01** |  | **ATKD** | **72.70** |

vanilla KD and ATKD. It is worth noting that ATKD provides comparable performance to KD's final performance after first 30th epoch training.

## 3.2 Performance Degradation Experiments

**CFIFAR-100** On CIFAR-100 task, We trained the ResNet14 with multiple teachers on the CIFAR-100 dataset, and the result is shown in Table 5. Experimental details follows the above settings. Under ATKD, student performance continues to increase as the teacher gets bigger. The training loss of ATKD is also much lower than the vanilla KD. In addition, the $G_{s\_gap}$ (computed with temperatures) metric shows that ATKD significantly reduces the sharpness gap.

In the analysis of section 2.3, we concluded that larger temperatures would have smaller sharpness gap. Therefore the comparison between KD and ATKD need to take temperature values into account. The temperature of KD is 4, while most of the mean adaptive temperatures used by ATKD are between $(3, 4)$. We provide more details in the appendix.

**ImageNet** Table 6 shows the degradation problem in ImageNet with ResNet18 as the student, and '-' denote that this specific experiment was not conducted in the cited paper. We compared ATKD with two previous methods that aim to alleviate the degradation problem, Early Stop (Cho & Hariharan, 2019) (ES) and Teacher Assistant (Mirzadeh et al., 2019) (TA). Both of these two methods explicitly regularizing the teacher capacity: 1) TA proposed to distill the large teacher to an intermediate teacher and then distill to the student, so that each knowledge distillation step has a better match between student and teacher capacity. 2) ES methods use the early stopped teacher, the teacher capacity would be regularized by fewer training steps.

We can see that ATKD exceeds Early Stop and TA methods with a large margin in all teacher settings. For example, when distilled by ResNet50 and ResNet152, the performance exceeds other methods by 2%. We obtained 73.01% accuracy, which is the best ResNet18 results as we know.

An interesting finding of the early stopped teacher method is that these regularized teachers have smaller $S_{sharpness}$, which can reduce the sharpness gap between the teacher and the student. The results is in Table 7. Similarly, the Teacher Assistant method also reduces the sharpness gap by using medium-sized teachers. This finding indicates that $S_{sharpness}$ metric could provide a universal framework to mitigate the degradation problem.

Table 7: The $S_{sharpness}$ of Early Stopped models, temperature set to 1.

| Network | ResNet14 | ResNet20 | ResNet32 | ResNet44 | ResNet56 | ResNet110 |
|---|---|---|---|---|---|---|
| $S_{sharpness}$ (240 epochs) | 12.20 | 13.11 | 13.84 | 14.45 | 15.37 | 16.13 |
| $S_{sharpness}$ (60 epochs) | - | 13.03 | 13.68 | 14.29 | 14.77 | 15.69 |

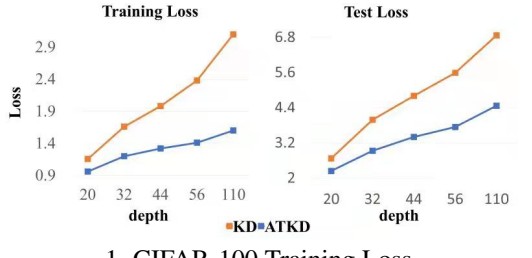

1. CIFAR-100 Training Loss

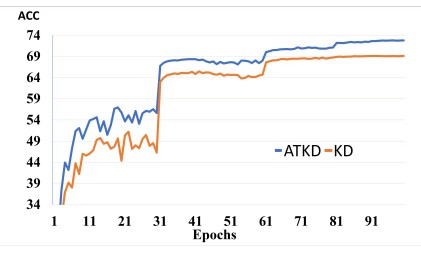

2. ImageNet Test Accuracy

Figure 1: 1. The training/test loss on CIFAR-100. With the teacher size grows, the loss of ATKD increases slower than KD, which shows that ATKD alleviates the Degradation Performance Problem. 2. The training process of KD and ATKD on ImageNet. ATKD achieves comparable accuracy with KD at the 30th epoch.

# 4 RELATED WORK

**Knowledge Distillation** Buciluă et al. (2006) first proposed compressing a trained cumbersome model into a smaller model by matching the logits between them. Then Hinton et al. (2015) advanced this idea and formed a more widely used framework known as knowledge distillation (KD). Knowledge distillation tries to minimize the KL divergence between the soft output probabilities generated by the logits through softmax. Xu et al. (2020) proposed to normalize the feature, the penultimate layer of the network, to perform distillation. This method is similar to our methods except that we perform on the logits layer. Knowledge distillation is also a kind of soft label training method. Similar to label smoothing, previous studies have found that knowledge distillation helps to regularize the training of network.

**Performance Degradation Problem:** Although distillation has shown a great potential in many tasks, a mysterious problem is found that larger teachers often harm the distillation performance, despite its more powerful ability (Cho & Hariharan, 2019; Mirzadeh et al., 2019). This problem is particularly severe on ImageNet, resulting in poor performance for KD. It was widely accepted that the mismatch of capacity caused this problem. Previous research proposed to regularize the teacher capacity to alleviate this problem heuristically. Cho & Hariharan (2019) proposed to early stop the training of the teacher. Moreover, Mirzadeh et al. (2019) proposed to use a medium-size teacher assistant (TA) to perform a sort of sequence distillation. This TA first learns from the teacher, then the student can learn from this TA. However, the accuracy of the early stopped teacher or TA is also harmed, which is lower than the original teacher.

# 5 CONCLUSION

The vanilla knowledge distillation overlooks the sharpness gap between the teacher and the student, which may cause the performance degradation problem. In this paper, we propose to use a metric to measure the sharpness of neural networks, allowing us to control the sharpness of models by using adaptive temperatures. Recently some papers have focused on combining label smoothing and knowledge distillation to train neural networks (Shen et al., 2021; Müller et al., 2019). An appealing point of view is that ATKD is inherently consistent with these methods. ATKD smooths teachers by adapting temperatures, while these methods let students learn from teachers trained by label smoothing. The deeper relationship between knowledge distillation and label smoothing remains to be explored.

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

## A APPENDIX

### A.1 EXPERIMENTS OF DIFFERENT FIXED TEMPERATURES

We conducted experiments of different fixed temperatures for vanilla KD (Table 8), which shows no observable performance improvement. We distilled ResNet14 by ResNet56 on CIFAR-100 dataset, and the rest details followed the setting in section 3.

### A.2 EXPERIMENTS VARIES WITH WIDTH

We add degradation problem experiments where teacher models vary with width here (Table 9). We use Wide ResNet to train on the CIFAR-100 dataset, the rest experiment details follow the setting in section 3.

Table 8: Experiments of different fixed temperatures

| $\tau^T$ | 4 | 4 | 4 | 4 | 4 | ATKD |
|---|---|---|---|---|---|---|
| $\tau^S$ | 3.5 | 3.7 | 4 | 4.3 | 4.5 | ATKD |
| Accuracy | 68.27 | 68.11 | 68.06 | 68.22 | 68.30 | 69.0 |
| $S_{s\_gap}$ | 0.23 | 0.25 | 0.30 | 0.35 | 0.37 | 0.19 |
| $\tau^T$ | 3.5 | 3.7 | 4 | 4.3 | 4.5 | 5 |
| $\tau^S$ | 4 | 4 | 4 | 4 | 4 | 4 |
| Accuracy | 67.79 | 68.01 | 68.06 | 68.10 | 68.07 | 68.01 |
| $S_{s\_gap}$ | 0.40 | 0.33 | 0.30 | 0.27 | 0.24 | 0.21 |

Table 9: Experiments varies with width

| | | WRN-16-2 | WRN-16-3 | WRN-16-4 | WRN-16-5 | WRN-16-6 |
|---|---|---|---|---|---|---|
| Test acc | Vanilla KD | 68.22 | 67.88 | 68.27 | 67.80 | 67.2 |
| | ATKD | **69.39** | **69.33** | **69.39** | **69.40** | **69.21** |
| $G_{s\_gap}$ | Vanilla KD | 0.37 | 0.49 | 0.59 | 0.71 | 0.83 |
| | ATKD | **0.10** | **0.22** | **0.34** | **0.41** | **0.55** |

## A.3 ENTROPY SCORE OF MODELS

Entropy was used as confidence score at Pereyra et al. (2017), that is related to the sharpness of model output. We measured the entropy values of different models (Table 10). Experimental details follow the setting in section 3.

Table 10: Logits Entropy

| ResNet14 | ResNet20 | ResNet32 | ResNet44 | ResNet56 | ResNet110 |
|---|---|---|---|---|---|
| 0.97 | 0.74 | 0.45 | 0.35 | 0.22 | 0.09 |

## A.4 SHARPNESS GAP DURING TRAINING

Figure 2 shows the sharpness gap changes during the training. ResNet20 and WRN-16-1 are used as students, and ResNet56 and WRN-16-3 are used as teachers respectively.

## A.5 EXPERIMENT ON SVHN

We add another dataset SVHN here (Goodfellow et al., 2013). Results show in table 11. SVHN is similar to MNIST (e.g., the images are of small cropped digits), but comes from real house numbers in Google Street View images. We use ResNet14 as the student. The rest experiment settings use the setting of CIFAR-100 in section 3. Further exploration on parameter tuning might get better results.

## A.6 THE SUM OF LOGITS

In the paper of Hinton et al. (2015), they presume that the $\sum_j z_j$ equals to zero during training without further analysis (In page 3, Eq. 3). Here, we analyze this assumption theoretically and experimentally.

It is well known that the gradient to logits $z_j$ is:

$$\frac{\partial L}{\partial z_j} = p_j - q_j \tag{16}$$

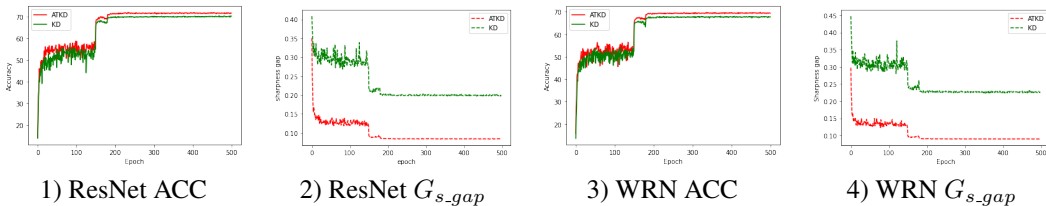

| 1) ResNet ACC | 2) ResNet $G_{s\_gap}$ | 3) WRN ACC | 4) WRN $G_{s\_gap}$ |

Figure 2: 1) The ResNet20 accuracy plot during training 2) The ResNet20 sharpness gap plot during training. 3) The WRN-16-1 accuracy plot during training 4) The WRN-16-1 sharpness gap plot during training

Table 11: Experiments on SVHN

|  |  | ResNet20 | ResNet32 | ResNet44 | ResNet56 |
|---|---|---|---|---|---|
| Teacher acc |  | 96.40 | 96.68 | 96.73 | 96.89 |
| Test acc | Vanilla KD | 96.57 | 96.54 | 96.61 | 96.59 |
|  | ATKD | **96.70** | **96.83** | **96.73** | **96.77** |
| $G_{s\_gap}$ | Vanilla KD | 0.05 | 0.05 | 0.07 | 0.07 |
|  | ATKD | **0.03** | **0.04** | **0.04** | **0.04** |

Where $p_j$ is the jth class probability of the student and the $q_j$ is the probability of the teacher. We can get the gradient to $\sum z_j$ by adding these gradients:

$$\sum_j \frac{\partial L}{\partial z_j} = \sum_j (p_j - q_j) = 1 - 1 = 0 \tag{17}$$

Therefore, the gradient to the sum of logits is zero.

Besides, we know that the logits is the product of the weight $W$ and the feature vector $f$ in the penultimate layer:

$$z = Wf = \sum_i W_i * f_i \tag{18}$$

Where $W_i$ is the ith column vector of $W$, considering that the initial weights of $W$ are sampled from the zero-meaned normal distribution in the popular initialization settings (He et al., 2015), we assume that the initialized $\sum_j W_{ij}$ (the sum of column vector) equals to zero. Therefore, the $\sum z_j$ should be zero all the time given the above assumptions.

We conducted experiments to verify this assumption. We train these models on the CIFAR-100 dataset. Table 12 shows that all these models logits are close to zero.

Table 12: The value of $\sum z_j$

|  | ResNet20 | ResNet32 | ResNet44 | ResNet56 | ResNet110 |
|---|---|---|---|---|---|
| Logits Sum | -5e-5 | -4.7e-5 | -5.7e-5 | -7.9e-4 | -6.1e-5 |

|  | WRN-16-1 | WRN-16-2 | WRN-16-3 | WRN-16-4 | VGG13 |
|---|---|---|---|---|---|
| Logits Sum | -4.8e-5 | -6.1e-6 | -5.5e-5 | -1.2e-5 | -3.1e-5 |

Table 13: Mean Value of Adaptive Temperatures

| ResNet14 | ResNet20 | ResNet32 | ResNet44 | ResNet56 | ResNet110 |
|---|---|---|---|---|---|
| 3.34 | 3.43 | 3.49 | 3.58 | 3.71 | 3.87 |

## A.7   THE ADAPTIVE TEMPERATURES

We provide the mean value of the temperatures in table 13.

