# OpenReview forum: "Reducing the Teacher-Student Gap via Adaptive Temperatures"
_ICLR.cc/2022/Conference — ICLR 2022 Submitted_

### Official Review · Reviewer_iJMG · 2021-10-30

**Correctness:** 2
**Technical Novelty And Significance:** 2
**Empirical Novelty And Significance:** 2
**Recommendation:** 5
**Confidence:** 4

**Details Of Ethics Concerns:**

There is no ethics concern mentioned in this paper. I think there is no ethic concern dataset and method proposed.

**Main Review:**

strengths

This paper attempted to improve the vanilla knowledge distillation method's performance from the fixed temperature side. Intuitively, this solution is reasonable and practical.

weaknesses

1. This paper is poorly written. Equations are confusing without the description of each component. For example, i and j in Eq(2). L_{KD} loss is also wrongly described. Also, the tables and captions are very unclear. For example, table1, which metrics are used? Top1 or Top5. In Table 6, what does '-' represent? What is TS? Figure 1, what is SKD?

2. Table 4 is supposed to be resnet18 as student and resnet34 as a teacher but ATKD with 73.01 is from teacher resent50. Check table 6.


**Summary Of The Paper:**

This paper is to improve the vanilla knowledge distillation method. Based on the observation in Cho&Hariharan that student degrades by oversized teachers, this paper proposed an adaptive temperature solution. A new metric called sharpness is introduced to quantify the teacher-student gap. Experiments are done on CIFAR100 and ImageNet.

**Summary Of The Review:**

The intuition of this paper is clear but the method proposed is not well presented, making understanding this work is hard.

---

> ### Author Response · Authors · 2021-11-23
> **Responses to the concerns**
>
> We sincerely apologize for the inconvenience. We have corrected these typos and other unclear parts of the paper (e.g., Eq(8) has been modified). We will continue to do our best to revise the manuscript to make it ready for publication.
>
> **A1:**
> 1. i and j denote the ith and jth value of logits. We have added descriptions to i and j in Eq(2) in the paper.
>
> 2. The $L_{KD}$ loss in Eq(2) is consistent with the many previous papers. (Cho \& Hariharan, 2019, Fu et al., 2020). If ''wrongly described'' means the cross entropy loss instead of KL divergence loss, the loss function of distillation can be either cross entropy or KL divergence, which is equivalent under the gradient descent optimization given the teacher network is fixed during distillation.
>
> 3. The metric used in Table 1 is top1 accuracy. The caption has been modified in Table 1.
>
> 4. The '-' means the results are missing in the cited paper. The description has been added on page 7.
>
> 5. The 'TS' is a typo for TA and SKD is a typo for ATKD. They have been fixed in our new version.
>
>
> **A2:**
>
>  Sincerely apologize for the inconvenience. We have corrected the results in the paper and added descriptions on page 6.

---

### Official Review · Reviewer_Yenx · 2021-11-02

**Correctness:** 3
**Technical Novelty And Significance:** 3
**Empirical Novelty And Significance:** 3
**Recommendation:** 6
**Confidence:** 4

**Main Review:**

Strengths:
- The proposed method can effectively reduce the sharpness gap between the teacher model and the student model and achieve better performance when a large teacher model is used.
- The work reveals a new perspective to investigate knowledge distillation.

Weakness:
- In table 5, please clarify that what temperature is used during training and computing the sharpness gap in the table. It's not fair if the sharpness is compared with different temperatures.
- Please explain in more detail that why the sum of student logits is a constant during training even if the input and parameters are zero-mean initialized.
- The logits 'std' defined on page 4 should be variance. And, why the mean of the output is assumed to be 0?



**Summary Of The Paper:**

The paper proposes to mitigate the performance degradation by controlling the sharpness gap between a large teacher and a student model. The sharpness is defined as the real softmax function (the logarithm of the sum of the exponentials of the logits). During the training, the temperature is set according to the sharpness of the logits. For the teacher model which has a sharp output, the temperature is larger so that the output becomes more smooth.


**Summary Of The Review:**

In summary, the paper proposes an interesting perspective to investigate the performance degradation in knowledge distillation using a large teacher model. However, some detail and assumption are not well-presented in the paper.

-- post-rebuttal

The response addressed some of my concerns. I will keep my score.

---

> ### Author Response · Authors · 2021-11-23
> **Responses to the concerns**
>
> Thank you for your efforts in reviewing our paper and providing helpful comments and suggestions.
>
> **A1:**
>
> The temperature used to compute the sharpness gap is the same as the temperatures used during training. The temperature of KD is 4, which is the default setting in many previous papers (e.g., Cho & Hariharan, 2019; Tian et al., 2020). Meanwhile, We count the adaptive temperatures and find that around 85% are smaller than 4 (teacher and student). We modified the paper on page 7 and added more results in appendix A.7.
>
> **A2:**
>
> We follow the assumption from Hinton et al., 2015 (On page 3, Eq. 3) that logits have been zero-meaned separately for each training example. However, the assumption lacked further analysis in the original paper. We have modified the paper on page 4 and added more analysis in appendix A.6. We provide a simple analysis here:
>
> 1. It is well known that the gradient of $z_i$ to entropy loss is $p_i -q_i$ (Hinton et al., 2015), where $p_i$ is the probability of student and $q_i$ is the probability output of teacher. Therefore the gradient to $\sum z_i$ is $\sum (p_i - q_i) = \sum q_i - \sum p_i = 1-1 = 0$, therefore we presume that the sum of logits is a constant during training.
>
> 2. Experimentally, we have verified that the sum of logits of the neural networks (regardless of model architecture and capacity) are very small values close to zero (like -5e-5, more results are in table 12).
>
> 3. In addition, considering the translation invariance of softmax, we can easily convert logits to zero-meaned vectors, so our analysis on page 3 is not strictly dependent on this assumption.
>
> **A3:**
>
> 1. We have corrected the 'std' typo and modified Eq.8 in our new version.
>
> 2. The explanation is the same as A2, and more details can be found in appendix A.6 of the modified paper. Experimental results are also provided in Table 12.

---

> > ### Comment · Reviewer_Yenx · 2021-11-28
> > **Responses to authors**
> >
> > Thanks for the clarification and analysis.
> >
> > A1. I assume the temperature used for ATKD is the adaptive one defined in Eq. 10. In that case, 1) Is it reasonable to conclude that the sharpness is decreased if ATKD and KD use different temperatures to compute the sharpness gap? 2) Is the sharpness gap equal to 0 if we substitute temperatures in Eq. 10 to Eq. 9?

---

### Official Review · Reviewer_6R6z · 2021-11-03

**Correctness:** 3
**Technical Novelty And Significance:** 3
**Empirical Novelty And Significance:** 3
**Recommendation:** 6
**Confidence:** 4

**Main Review:**

Strengths

The authors:

(1) Provide a new perspective to explain the degradation problem of KD.

(2) Introduce the realsoftmax function to reflect the sharpness of models, and propose ATKD to adaptively narrow the sharpness gap for the steady knowledge distillation.

Weaknesses

The authors should provide more analysis:

(1) What is the best sharpness gap between the teacher model and the student model? A theoretical analysis?

(2) Can you manually adjust the temperature of the teacher (fix the temperature of the student) and report the performance changes of the student? It would be better to contrast the sharpness gap of the best student with that of ATKD.

(3) Whether the best sharpness gap varies on different datasets or tasks?


**Summary Of The Paper:**

This work aims to alleviate the performance degradation problem of the student model during knowledge distillation. The authors argue that the degradation problem may come from the sharpness gap of the model outputs.

They first demonstrate the relation of the sharpness gap (between teacher and student) and the degradation degree, then introduce the realsoftmax function to measure the sharpness of the model output, and finally propose ATKD to adaptively change the temperatures of the teacher and the student for reducing the sharpness gap.

Besides, the authors also analyze previous methods (e.g., Early Stop and TS) from the perspective of the sharpness of models.


**Summary Of The Review:**

The idea is novelty and interesting. The authors should provide more analysis to demonstrate the effectiveness of this method comprehensively.

I would recommend this paper for ICLR2022.

---

> ### Author Response · Authors · 2021-11-23
> **Responses to the concerns**
>
>
> Thank you for your efforts in reviewing our paper and providing helpful comments and suggestions.
>
> **A1:**
>
> Since the sharpness gap hinders the learning of the student model from the teacher model, the best sharpness gap should be zero. Under second-order approximation, ATKD can effectively reduce the sharpness gap and improve performance.
> Besides ATKD, one way to reduce the sharpness gap is to use a higher fixed teacher temperature.  We discuss this situation below (A2).
>
> **A2:**
>
> 1. Thanks for the advice, and we have updated the results in table 8 of appendix A.1.
>
> 2. Table 8 shows that increasing the temperature of the teacher model reduces the sharpness gap and improves student performance by a very modest margin. In Table 8, the student gets the best result when the teacher temperature is set to 4.3, where the sharpness gap is still larger than the result of ATKD.
>
> 3. Follow the assumption of Hinton et al., 2015, if the teacher temperature is very high, the student could pay more attention to the noisy values in teacher logits, which could counteract the benefit from the smaller sharpness gap.  Therefore, it is important that ATKD has a fast sharpness gap convergence rate, which avoids inducing too much noise.
>
> **A3**
>
> Like the A1, the best sharpness gap should be zero.
> We conducted experiments on SVHN with the vanilla KD and ATKD.  The result can be found in Tabel 11 of appendix A.5. The result is consistent with the observation of CIFAR-100. ATKD performance better than KD and achieves a lower sharpness gap.

---

> > ### Comment · Reviewer_6R6z · 2021-11-29
> > **Responses to the authors: concerns for the best sharpness gap and its relation with performance**
> >
> > Thank you for the detailed reply.
> >
> > According to the results in Table 8, it is obvious that the sharpness gap can be large even the performance gets significant improvement.
> > Therefore, there is no convincing evidence to verify the assumption that the best sharpness gap should be zero.
> >
> > ### For Q1, the authors do not provide a related answer.
> >
> > ### For Q3, the authors also do not address my concern.

---

> > > ### Author Response · Authors · 2021-11-30
> > > **Responses the concerns for the best sharpness gap and its relation with performance**
> > >
> > > One of our main contributions in this work is that we found a smaller sharpness gap could lead to better student performance while keeping other factors the same. Table 8 does not show significant results because the student performance is also affected by other factors (i.e., the noise contained in the teacher logits). On the other hand, the comparison between KD and ATKD results in Table 5, Table 9, and Table 11 (compared vertically) show that without the interference from other factors, smaller sharpness gaps lead to better student performance. We provide the theoretical analysis below:
> > >
> > > **1. The sharpness gap is affected by the teacher capacity and temperature**
> > >
> > > The sharpness gap is:
> > > \begin{equation}
> > >     G_{s-gap} = \log\sum_{j} e^{v_{j} / \tau} - \log\sum_{j} e^{z_{j} / \tau}
> > > \end{equation}
> > >
> > > Given a student model, it shows that the temperature and the teacher model affect the sharpness gap.
> > > In addition, the temperature and the teacher capacity can also affect the teacher logits noise (the negative values in teacher logits) and the teacher accuracy, respectively.
> > > These three factors, sharpness gap, teacher accuracy, and the teacher logits noise,  finally affected the student performance.
> > >
> > > **2. Change the teacher capacity**
> > >
> > > The accuracy of the student shows an inverted V curve when changing the teacher capacity, where the best sharpness gap would be somewhere in the middle (the vanilla KD result in Table 5, Table 9). The teacher accuracy and the sharpness gap are two main factors here. Suppose the temperature is fixed, and we decrease the teacher model capacity from ResNet110 to ResNet20 (Table 5), the sharpness gap and the teacher accuracy would decrease together. In that case, the best sharpness gap would be the one where the negative impact of the decreased teacher accuracy and the positive impact of the decreased sharpness gap has reached a balance. It can be seen from Table 5 that the student distilled by ResNet32 gets the best performance in KD. In the experiment of Table 9 (Wide ResNet), similar phenomenons can be observed. The result of Table 11 (SVHN) shows a slightly different pattern, where the distillation result by teacher ResNet20 (96.57) and ResNet44(96.61) are both better than that of teacher ResNet32 (96.54). This may be because that these teachers share similar accuracy (ResNet20 96.40, ResNet32, 96.68, ResNet44 96.73) and sharpness gap (ResNet20 0.05, ResNet32 0.05, ResNet44 0.07, decimals are truncated) in this dataset. The unadjusted hyperparameters inherited from CIFAR-100 may be another reason. We will continue to conduct more experiments on SVHN and add more results on other datasets like ImageNet and CIFAR10.
> > >
> > >
> > > **3. Change the teacher temperature**
> > >
> > > Suppose the teacher model is fixed, and we increase the teacher temperature.
> > > Although higher temperatures can reduce the sharpness gap,  the student could pay too much attention to the negative values in the teacher logits, and these negative values could be very noisy (Hinton et al., 2015). Under very high temperatures, the student model would learn almost uniform distribution, which would harm the student performance. It can be seen from the second row of Table 8 that increased teacher temperature does not bring considerable performance improvement.
> > >
> > > **4. The comparison between KD and ATKD**
> > >
> > > a ) When distilled by the same teacher, the comparison between ATKD and KD results In Table 5, Table 9, and Table 11 (compared vertically) avoids the interference brought by the changed teacher accuracy. In addition, the student would not pay too much attention to the negative noise logits because the teacher's adaptive temperatures are close to the student's temperatures (Table 13). Therefore, the reduced sharpness gap brings better performance for ATKD.
> > >
> > > b ) When changing the teacher capacity, the accuracy of ATKD shows a similar inverted V curve like KD in Table 9, but the best result of ATKD is with teacher WRN-16-5, while KD is with WRN-16-4. Moreover, in Table 5, the best sharpness gap is obtained with the largest teacher ResNet110.
> > > This can be explained that the reduced sharpness gap of ATKD (compared to KD) allows students to learn from larger and more accurate teachers and be improved (naturally larger teachers bring larger sharpness gaps regardless of the KD method).
> > >
> > > Finally, thank you for your effort in reviewing our paper. We will revise our paper in the future version.

---

### Official Review · Reviewer_iBXp · 2021-11-03

**Correctness:** 3
**Technical Novelty And Significance:** 3
**Empirical Novelty And Significance:** 3
**Recommendation:** 6
**Confidence:** 4

**Main Review:**

The sharpness gap analysis and observations are interesting. It shows a strong correlation between the sharpness gap and KD results, providing new insight into the KD strategy. The proposed ATKD method is simple and has a reasonable theoretical justification. Its result achieves SOTA performance, especially the large gain in the ImageNet experiment (Tables 4 and 6); such a performance gain is non-trivial and considered significant.

This work raised several interesting questions. Some of them probably should be answered in the paper to conclude the observations about sharpness better:
1. Would other sharpness functions/scores lead to the same conclusion? For example, would the entropy of output probability a reasonable sharpness score, and how does it compare to the realsoftmax function?
2. Any guess about why a student with vanilla KD loss can not learn to match the sharpness of the teacher? The limited capacity of the student may not be a candidate cause since the student model can multiply a large number to the logits to make the sharpness score large, thus reducing the sharpness gap.
3. Section 2.3 suggests that the classical KD loss converges slower than the proposed ATKD loss. This indicates that the classical KD converges slower but does not have a worse error bound. However, in practice, would training longer with the classical KD loss converge to a similar sharpness gap and classification performance obtained by ATKD?
4. Would the proposed sharpness score and observation generalize to other cases? The observations in the paper are mainly based on varying the depth of the model. Does the same trend hold when changing the width of the teacher model?
5. One additional observation that could be made is to have a line plot that has training epochs in the x-axis and has the sharpness gap and the student classification accuracy in the y-axis.


**Summary Of The Paper:**

This work explores how the sharpness gap between a teacher model and a student model affects knowledge distillation performance. The paper argues that a large sharpness gap harms KD and proposes an adaptive temperature strategy to speed up the decrease of the sharpness gap in training. The experiments demonstrate a high correlation between the sharpness gap and student performance. It also shows that the proposed adaptive temperature strategy can achieve SOTA results on the image classification tasks.

**Summary Of The Review:**

The work provides an interesting insight into KD by investigating the sharpness gap between the teacher and student model. The ATKD method built based on this insight is simple but can significantly improve the KD results. However, this work still leaves several important questions around the justification of sharpness discussed above. The mixed pros and cons lead to my current rating, subject to be updated.

---

> ### Author Response · Authors · 2021-11-23
> **Responses to the concerns**
>
> Thank you for your efforts in reviewing our paper and providing helpful comments and suggestions.
>
> **A1:**
>
> Thank you for your suggestion. Pereyra et al., 2017 used entropy to regularize the confidence of the network, which is similar to the sharpness defined here. We measured the entropy of various models and found that the entropy would become smaller as the model becomes larger, which is consistent with our conclusion that larger models become sharper. The results are as follows:
>
> | ResNet14 | ResNet20 | ResNet32 | ResNet44 | ResNet56 | ResNet110 |
> |----------|----------|----------|----------|----------|-----------|
> | 0.97     | 0.74     | 0.45     | 0.35     | 0.22     | 0.09      |
>
> **A2:**
>
> One possible candidate is that the gap between the teacher model and the student model is inconsistent across different samples.
> 1. According to (Cho \& Hariharan, 2019), because of the capacity gap, the predictions of the student model and the teacher model for many samples are inconsistent (page 6). If the student model multiplies with a larger number, although it may reduce the loss on the samples where the student and the teacher predict the same, it would increase the loss on those samples with inconsistent predictions.
>     For example, on a binary classification task, if the probability distribution of the student model is (0.3, 0.7) and the teacher model is (0.9, 0.1), multiplying the model output with a larger number would increase the distillation loss.
>
> 2. In addition, for those samples where students and teachers predict the same, multiplying the student model by a larger number may also increase the loss, because the sharpness gap between the samples may not be the same.
> 	For example, also in the binary classification task, if the teacher model's prediction probability distributions for two samples A and B are (0.9,0.1) and (0.6,0.4) respectively, while the student model's predictions are (0.8,0.2) and (0.6,0.4). At this time, multiplying the student model by a number will make the student model's predictions to be (0.9,0.1) and (0.65,0.34), which also increases the loss. Therefore, it is non-trivial to multiple a single number to reduce the sharpness gap.
>
> **A3:**
>
> 1. Section 2.3 discusses the relationship between the sharpness gap and temperatures. At extremely high temperatures, KD and ATKD indeed have the same lower bound, but in practice, the temperature is generally small values like 4 (Hinton et al., 2015, Zehao Huang et al., 2017, Sergey Zagoruyko et al., 2017) Hinton et al., 2015 analyzed that high temperatures could harm the performance of distillation because the student model would pay too much attention to those small negative values in the logits, and these small negative values could be very noisy (page 3). Therefore, in practice, the sharpness gap of ATKD will be lower than that of KD with normal temperatures.
>
> 2. Regarding the relationship between training time and sharpness gap, we conducted experiments and found that after a certain period of time, longer training time does not reduce the sharpness gap. For example, ResNet20 distilled by ResNet56 on CIFAR-100 achieved sharpness gap of 0.2015 at the 200th epoch and 0.2018 at the 500th epoch. Details can be found in Figure 2 of appendix A.4.
>
> **A4:**
>
> The result of experiments varies with width has been added in the appendix (Table 9 of A.2). The results show a similar trend to the depth experiment.
>
> **A5:**
>
> Thanks for the advice, the result is updated in the appendix (Figure 2 of A.4). The changes of accuracy and sharpness gap show certain synchronicity.
>
> Finally, thanks for your positive evaluation of our work. The only thing we can do now is to try our best to make our manuscript satisfy the level of an ICLR paper, and we are looking forward to your further comments.

---

> > ### Comment · Reviewer_iBXp · 2021-11-24
> > **A4**
> >
> > Thank you for the additional experiments. I have a further question about these observations. Specifically, In Table 9, WRN-16-6-ATKD-$G_{s_gap}$ is 0.55 with an accuracy of 69.21, while WRN-16-2-Vanilla-$G_{s-gap}$ is 0.37 with accuracy 68.22. In other words, the $G_{s_gap}$ is larger, but the accuracy is better. The same phenomenon appears in Table 5 as well. How to interpret the results?
> >
> > BTW, could you please provide the teacher's accuracy of Table 9? Thanks!

---

> > > ### Author Response · Authors · 2021-11-29
> > > **Responses to the concerns**
> > >
> > > **A4:**
> > >
> > > **1:**
> > > The accuracy of the teacher is an important factor besides the sharpness gap.  For example, the accuracy rate of WRN-16-6 is 78.4, while the accuracy rate of WRN-16-2 is only 72.9. Similarly, in Table 5, the accuracy of ResNet110 is 74.31, while the accuracy of ResNet20 is only 69.40.
> > >
> > > **2:**
> > > Teacher accuracy:
> > >
> > > |             | WRN_16_2 | WRN_16_3 | WRN_16_4 | WRN_16_5 | WRN_16_6 |
> > > | :---------: | :------: | :------: | :------: | :------: | :------: |
> > > | Teacher Acc |   72.9   |   75.6   |   76.7   |   77.9   |   78.4   |

---

> > ### Comment · Reviewer_iBXp · 2021-11-24
> > **A1**
> >
> > This question asks replacing the realsoftmax of Equation 6 with entropy: $G_e=H(p^T)-H(p^S)$. Would $G_e$ show the same trend as $G_{s\-gap}$ in Table 5? Is there a justification for using realsoftmax rather than entropy?

---

### Author Response · Authors · 2021-11-23
**To all reviewers**

We thank all reviewers for their valuable comments. We are encouraged that the reviewers think our paper or our proposed method is simple and reasonable(\#R1,\#R4), novelty (\#R2), interesting (\#R1, \#R2, \#R3), effective (\#R2, \#R3), practical (\#R4) and provide new perspective to investigate knowledge distillation or performance degradation problem (\#R2, \#R3). While there are still some concerns or misunderstandings, we will try our best to resolve them for each reviewer.

---

### Decision · Program_Chairs · 2022-01-20

**Decision:**

Reject

**Comment:**

The authors study the degradation problem observed in KD for large teacher networks and propose to address it by quantifying and adapting to a *sharpness gap* between the student and the teacher. The reviewers generally appreciated the proposed approach in handling larger teachers and found it effective within the scope of the numerical results provided in the paper. That said, the reviewers raised several critical issues concerning the writing and the presentation of several crucial parts of the paper, in particular those related to the sharpness measure and the proposed training method ATKD. Thus, given this, and the exchanges between the reviewers and the authors, in its present form, the paper cannot be recommended for acceptance. The authors are encouraged to incorporate the valuable feedback provided by the knowledgeable reviewers.